

**Analysis of geomagnetic measurements prior the Maule (2010), Iquique (2014) and Illapel (2015)**
**earthquakes, in the Pacific Ocean sector of the Southern Hemisphere.**
Enrique G. Cordaro (1, 2), Patricio Venegas-Aravena (1, 3, 4) and David Laroze (5, 6).
(1) Observatorios de Radiación Cósmica y Geomagnetismo. Dep. Física, F. C. F. M.  Universidad de Chile, Casilla 487-
3, Santiago, Chile.
(2) Facultad de Ingeniería. Universidad Autónoma de Chile. Pedro de Valdivia 425. Santiago. Chile
(3) Departamento de Geofísica Universidad de Chile, Blanco Encalada 2002, Santiago, Chile.
(4) Department of Structural and Geotechnical Engineering, School of Engineering, Pontificia Universidad Católica de
Chile, Vicuña Mackenna 4860, Macul, Santiago, Chile.
(5) Instituto de Alta Investigación, CEDENNA, Universidad de Tarapacá, Casilla 7D, Arica, Chile.
(6) School of Physical Sciences and Nanotechnology, Yachay Tech University, 00119 Urcuquí, Ecuador.
E. G. Cordaro: ecordaro@dfi.uchile.cl
P. Venegas- Aravena: patricio.venegas@ing.uchile.cl
D. Laroze: dlarozen@uta.cl
**ABSTRACT**
It has been possible to detect variations in the vertical component of the geomagnetic field (Bz) through its first and
second derivate in a range of frequencies (μHz); these seem to be roughly related with some major seismic subduction
events. We studied the period 2010-2015, analysing the daily values of magnetic records over periods close to the
last three significant events that occurred through the Chilean margin, i. e., along a boundary between convergent
plates that is characterized by the occurrence of seismic events of magnitude greater than Mw8. These are the events
of Iquique 2014, Illapel 2015 and Maule 2010, all at different latitudes, on different dates and characterized by
different types of margin (erosive or accretionary). Certain similarities were found in the associated magnetic field
variations: 1) Variation in the radial or z component of the geomagnetic field and its first and second temporal
derivative, modelled as a small jump, and small oscillations in the second derivative, generating a frequency band
between 1c / 48.9 hours and 1c / 79.13 Hrs. 2) A variable time lapse of between 30 and 120 days; and 3) The seismic
event. Furthermore, when analysing spectrograms for the second temporal derivate of the radial component,
different behaviour is found related to its spectral density. This takes the form of an increase in ultra-low frequencies
(0.01-0.4 mHz) between the start of the magnetic jump and the seismic event. These frequencies are lower than those
found during the last years by research groups that related magnetic field and earthquakes, furthermore the concept
of time lapse close to 30 days is in agreement with those research groups. The previous analyses may not be so robust,
this is why additionally a new method is used with stations closer to the events and time periods of two years. We
analysed the daily cumulative number of anomalous behaviour in z component of magnetic field on ground based
magnetometers. The results show an increase in the number of magnetic anomalies prior to the occurrence of the
three earthquakes. The behavior of the anomalies is similar to those presented by other authors for other
earthquakes with similar methods in ionosphere. All this magnetic features might recover seismic information of the
events and could be related with Lithosphere-Atmosphere-Ionosphere Coupling.
Keywords: Geomagnetism, South Atlantic Magnetic Anomaly, Lithosphere-Atmosphere-Ionosphere Coupling,
Tectonic plates, Earthquakes

**1.-Introduction.**
The object of this paper is to show the most significant characteristics of the magnetic field and its possible relation
with the last three Chilean earthquakes. Specifically, we used the $B_z$ component recorded in the Putre (OP), Easter or
Pascua Island (IPM), Osorno (OSO) and Pilar (PIL) observatories. Cordaro et al., 2018, 2019 showed that the variations
of the geomagnetic cutoff rigidity (magnetic shielding against the solar wind) have some relation with the Chilean
convergent margin. Furthermore, Cordaro et al. (2018) showed that the frequencies of the micro hertz order in the
vertical magnetic field also could have some relationship with the earthquake of Maule 2010 Mw8.8. However,
Vallianatos and Tzanis (2003) showed that the magnetic field frequencies possibly related to earthquakes comprise a
range of at least three orders of magnitude. The last researches have corroborated that the magnetic coupling of
ionosphere-lithosphere-atmosphere is statistically related to some earthquake events through variations in the earth
magnetic field detected for one month before the seismic events in the  frequency range of 5-100 mili-Hertz
[Hayakawa and Molchanov, 2002, Pulinets and Boyarchuk, 2004, Varotsos, 2005, Balasis and Mandea 2007,
Molchanov and Hayakawa, 2008, Liu, 2010, Hayakawa et al., 2015, Contoyiannis et al., 2016, Potirakis et al., 2016, De
Santis et al., 2017, Oikonomou et al., 2017, Marchetti and Akhoondzadeh, 2018, Potirakis et al., 2018] fulfilling the



three orders of magnitude in the frequency of the magnetic field proposed by Vallianatos and Tzanis (2003). In consequence, in this paper we focus in the geomagnetic activity before three of strongest earthquakes that hit the Chilean margin during the last years: Maule 2010 (Mw8.8), Iquique 2014 (Mw8.2) and Illapel 2015 (Mw8.3).

For other hands, events on the Chilean margin of magnitude greater than Mw8 are generally associated with subduction events, while smaller events are considered of minor interest, and could even be triggered by other events. An example is the event of March 11, 2010 (Mw7), which was a reactivation of the Pichilemu deep intra-plate fault [Farías et al., 2011]. Despite the specific detailed characteristics of the northern and southern parts of the Chilean margin, all events are triggered by small variations or disturbances in stress, allowing stress to fall and energy to be released [Ranjith and Rice, 1999, He et al., 2003].

Under these conditions, we will study the Chilean territory by examining the variations in the first and second derivative of the Bz component of the magnetic field and the variations in seismic movements produced in the region. We use lower frequencies, longer time series and methods that are different from the ones which have been previously used [Contoyiannis et al., 2016, Potirakis et al., 2016, De Santis et al., 2017, Oikonomou et al., 2017, Marchetti and Akhoondzadeh, 2018]. That is, the Fourier analysis in the range of micro hertz for daily averages values and the Spectrograms method in the range of mili Hertz per minute values. Conceptually, the second derivative of B indicates a change in secular acceleration, which we will call a *jump*. We also introduce the daily cumulative numbers of anomalous behaviour in the component z of magnetic field over the surface of Earth, in the lithosphere. Similar ideas were proposed by De Santis et al. (2017) and by Marchetti and Akhoondzadeh (2018) for Nepal 2015 Mw7.8 and Mexico 2017 Mw8.2 earthquakes respectively. In the method applied to the lithosphere we identify the magnetic records with less external disturbances, the variations of the surplus records are considered of lithospheric origin.

In section 2 we introduce some of the scientific efforts related to magnetism and seismic events. We were able to use the radial component of the magnetic field, its first derivative or secular variation and its second derivative or secular acceleration to try to answer the question as to the origin of the highest-energy seismic movements and look for a possible precursor in section 2.1. In order to do this, we developed an experimental procedure based on the geomagnetic data for three of the most intense seismic events of the South Pacific region: the 2010 Maule earthquake, the 2014 Iquique earthquake and the 2015 Illapel earthquake, all in Chile.

In section 3 we introduce some manifestation of Space climate in the geomagnetic field during the periods concerned is defined by the Kp magnetic activity index for the months previous to the three earthquakes: Maule 2010 (Dec 15, 2009 to Mar 15, 2010), Iquique 2014 (Jan 1, 2014 to Apr 15, 2014) and Illapel 2015 (Jul 1, 2015 to Sep 30, 2015).

In the section 4 we present an additional methodology using the daily cumulative number of Bz anomalies on Earth surface. Here we use different stations: OSO, with |Dst| <10 nT for the Maule earthquake, and PIL station with |Dst| < 5 nT for the Iquique and Illapel earthquakes. Using quiet times for a period of two years for each earthquake. Finally, we present discussion and conclusions in section 5.

## 2. Magnetic field and Seismology.
### 2.1. Observatories of geomagnetic data and Magneto-seismic relation.

The great majority of studies of the magnetic field and seismic events have been carried out relating temporal variations in the geomagnetic field with frequencies which are comparable to variations in the ionosphere, the space medium and human activity ($0.01 - 10\ Hz$). Besides, they have required specific chemical and geological conditions in fault zones, which restrains the arguments of such studies as they cannot establish any sort of causal mechanism [Thomas et al., 2009a, Thomas et al., 2009b, Thomas et al., 2012, Love and Thomas, 2013, Scoville et al., 2015]. We used the first and second derivatives of the $B_z$ component in the Putre and Ester Island (IPM) station magnetometers, which have not been thoroughly investigated.

A high correlation between the vertical component of the earth's magnetic field and seismic activity at the Putre station was found. We therefore seek to specify this behaviour in a shorter time window than the period studied previously (1955-2010). We consider the most significant events that have occurred in recent years in the network of observatories, given their proximity to the Chilean convergent margin and the quality of the data. We start with an analysis of the behaviour of the space medium and its influence on measurements.

We obtained the values of secular variations in the magnetic field from the Putre (OP), Los Cerrillos (OLC) and Easter Island (IPM) observatories. (Note that the IPM station was closed in 1968 and subsequently reactivated in 2008 by the French INTERMAGNET Group and the Meteorological Service of Chile) [Chulliat et al., 2009, Soloviev et al., 2012].



The Putre observatory (OP) is at 18°11'47.8S, 69°33'10.9W, 3,598 m.a.s.l (meters above sea level); Los Cerrillos (OLC)
is at 33°29'42.3S, 70°42'59.8W, 570 m.a.s.l. They are both located on the western edge of the South American Plate,
1,700 kilometres apart from each other. This zone includes the South Atlantic Magnetic Anomaly (SAMA), the centre
of which is 1,700 kilometres east of these two observatories, forming an equilateral triangle. IPM is located at 27.1°S,
109.2W, 82,83 m.a.s.l, on the western edge of the Nazca plate, characterised as a hotspot [Vezzoli and Acoocela,
2009]. OSO is located in the coordinates 40°20'24"S, 74°46'64"W and PIL at 31°40'00.0"S, 63°53'00.0"W.
In the Putre and Los Cerrillos observatories, a diminution in the values of the whole magnetic field and each of its
components is found. This can be attributed to the fact that the OLC and OP observatories are influenced by the South
Atlantic Magnetic Anomaly, while on Easter Island the influence of SAMA is weaker [Storini et al., 1999].
The scientific and technical characteristics of the Putre (OP) and Los Cerrillos (OLC) observatories, i.e. location,
altitude, atmospheric depth, type of detectors, geomagnetic cutoff rigidities and operating times, may be found in
[Cordaro et al., 2012, Cordaro et al., 2016] while for Easter Island (IPM) the information is available in SuperMag
Network [Chulliat et al., 2009, Gjerloev, 2012]. The main characteristics for the observatories as location, altitude,
atmospheric depth, type of detector, and operations time, are shown in Table 1.
The first significant property in the secular variation of the geomagnetic field is determined by the shielding it
exercises on the cosmic ray particles which reach the Earth, called the geomagnetic cutoff rigidity. It is defined as the
product of the force of the magnetic field and the curvature radius of the incident particle $r_g$, which relates the
geomagnetic latitude with the classification of the particle trajectories and could be related to geological features in
the Chilean margin [Pomerantz, 1971, Shea and Smart, 2001, Smart and Shea, 2005, Cordaro et al., 2018, Cordaro et
al., 2019].
The variations shown during the period 1955-2010 which can explain the non-performance of the latitude effect
[Pomerantz, 1971] are of internal origin [Bloxham, 2002, McFadden, 2007, Sarson, 2007, Finlay, 2007]. 3D models of
core mantle boundary (CMB) topology based on the velocities of seismic waves [Simmons et al., 2010] show the
existence of positive topography in upthrust regions and negative topography in subduction zones [Lassak et al., 2010,
Soldati et al., 2012, Yoshida, 2008].
It must also be remembered that the intensity of the geomagnetic field at the surface varies between 20,000 and
60,000 x $10^{-9}$ T (from the equatorial zone to the Antarctic in the Pacific Antarctic Sector of the southern hemisphere),
and that within the outer core it is estimated to be of the order of 2-4 mT (rms) [Olson, 1999, 2015], thus by focusing
attention on measurements of the geomagnetic field (B, B' and B''), evidence can be found of a possible pattern in
component $B_z$ in the highest-energy seismic events (Mw > 8.0) occurring in subduction zones during the period 2005-
2011, i.e.:. The pattern of the $B_z$ component consists of a jump of the order of 100-300 nT for B , its first temporal
change of B as $B'_z$ and its second temporal change of B as $B''_z$ and a time lapse between the jump and the seismic
event of the order of 30 to 120 days. This was accompanied by variations in the geomagnetic field with frequencies
in the range of 3 to 5 micro Hz, which are internal [Bloxham, 2002, McFadden, 2007, Sarson, 2007].
A most direct research that relates geomagnetic variations and earthquakes is performed by the variations in the
ultra low frequency (ULF) in the magnetic field due the ionosphere-lithosphere-atmosphere coupling [Hayakawa et
al., 2015, De Santis, 2015, Contoyiannis et al., 2016, Potirakis et al., 2016, De Santis et al., 2017, Oikonomou et al.,
2017, Marchetti and Akhoondzadeh, 2018]. The spatial coupling at the earth's surface is given by $r = 10^{0.43M}$ (M:
earthquakes magnitude), and it means the maximum radius from the epicentre where evidence of magnetic
perturbations due ionosphere-lithosphere-atmosphere coupling is expected to be found [Dobrovolsky et al., 1979 ;
Pulinets and Boyarchuk, 2004, Oikonomou et al., 2017]. Table 2 shows that all magnetic measurements performed at
the stations of Putre and IPM are inside, or at the boundary of each radius determined by the earthquakes magnitude
of Maule, Iquique and Illapel, thus it is expected that the magnetic measurements are influenced by the ionosphere-
lithosphere-atmosphere coupling.
**3.0 Space Climate and the Seismic events of 27/2/2010 in Maule, 1/4/2014 in Iquique and 16/9/2015 in Illapel.**
The manifestation of space climate in the geomagnetic field during the periods concerned is defined by the Kp
magnetic activity index as shown in [Figure 1 a, b, c] for the months previous to the three earthquakes: Maule 2010
(Dec 12, 2009 to Mar 15, 2010), Iquique 2014 (Jan 1, 2014 to Apr 15, 2014) and Illapel 2015 (Jul 1, 2015 to Sep 30,
2015). For Maule 2010 the magnetic activity reached a kp index equal to or greater than 4 on only three isolated
occasions, it is therefore considered a calm period; for Iquique 2014, activity was concentrated around Feb 19, 2014
while for Illapel 2015 the maximum activity was recorded between Sep 8 and 10. In all three cases, activity did not
persist in time. The magnetic records for the $B_z$ component show little external influence.



The magnetometer data in the OP and IPM stations for the Maule event run from Oct 31, 2009 to Apr 3, 2010; for
Iquique from Nov 15, 2013 to Apr 15, 2014 and for Illapel from Jul 1, 2015 to Sep 20, 2015. The Maule 2010 event had
a magnitude of Mw8.8 and occurred on Feb 27, 2010 at 06:34 UTC at a depth of 35 km, at 35.909°S, 72.733°W. Iquique
2014 had a magnitude of Mw8.2 and occurred on Apr 1, 2014 at 23:46 UTC at a depth of 25.0 km ± 1.8 km, at 19.610°S
70.769°W. Illapel 2015 had a magnitude of Mw8.3 and occurred on Sep 16, 2015 at 23:46 UTC at a depth of 22.4 km
± 3.2 km [USGS].
Measurements of the $B_z$ component [Figure 2] are represented by similar gradients in Iquique 2014 and Illapel 2015
to those found in Maule 2010, giving rise to a jump in each case. It is known that these magnetic signals are generated
by the earth's core and disseminated through the mantle, implying changes in its electrical conductivity [Stewart. et
al., 1995].
The jump in the $B_z$ component for Maule 2010 was recorded in the OP station on Jan 23, 2010, a time lapse of 36 days
before the earthquake and the moment at which a change appears in the gradient or trend. It alters from a diminution
of 225 nT in the period Oct 31, 2009 to Jan 23, 2010, to a less abrupt diminution of 30 nT between Jan 23, 2010 and
Apr 3, 2010; prior to the jump on Jan 16, 2010 there is a small, abrupt diminution from -5048 nT to -4927 nT.
Discounting this small, abrupt diminution, the delta between the gradients falls from -4960 nT to -4926 nT, delta = 34
nT [Figure 2 a]. For Iquique 2014 the jump recorded in OP occurred on Dec 27, 2013, a time lapse of 96 days before
the earthquake. A change appears in the gradient on this date from a diminution of 123 nT in the period Nov 15, 2013
to Dec 27, 2013, to a diminution of 113 nT between Dec 27, 2013 and Apr 15, 2014; the jump presents a change from
-7355 nT to -7235 nT, delta = 120 nT [Figure 2 b]. For Iquique 2014 the jump measured at IPM occurred on Apr 3,
2014, a time lapse of 91 days before the earthquake. The trend shows a slight increase between Sep 30, 2013 and Jan
3, 2014, from -19116 nT to -19104 nT, while a further slight increase occurs in the period Jan 3, 2014 to May 6, 2014,
from -19101 nT to -19099 nT. The size of the jump was -3 nT [Figure 2 c]. For Illapel 2015 the jump measured at IPM
occurred on Aug 31, 2015, a time lapse of 16 days before the earthquake. The trend shows a slight diminution
between Aug 31, 2015 and Sep 20, 2015, from -19054 nT to -19072 nT, a jump of -11 nT.
Note that the gradient measured from Easter Island is the reverse of the gradients in the Maule and Iquique regions
[Figure 2 c]; the gradient is gentler because IPM is located on the opposite edge of the Nazca plate.
For the first derivative (secular variation of the $B_z$ component) the maximum value (or peak value) observed for Maule
2010 was 141 nT/day on Jan 23, 2010.  Two peaks are found for Iquique 2014, one of 84 nT/day on Dec 5, 2013 and
another one of 78 nT/day on Dec 28, 2013; one peak is recorded for Illapel 2015 of 8.0 nT/day. For the second
derivative (secular acceleration of the $B_z$ component) a positive peak is recorded for Maule 2010 of 157 nT/day$^2$ on
Jan 22, 2010 and a negative peak of -116 nT/day$^2$ on Feb 24, 2010]. For Iquique 2014 a positive peak of 98 nT/day$^2$
was observed on Dec 29, 2013 and a negative peak of -127 nT/day$^2$ on Dec 31, 2013]. In Illapel 2015, during the days
immediately before the jump a negative peak of - 2 nT/day$^2$ was recorded on Aug 31, 2015 and a positive peak of 0.01
nT/day$^2$ on Sep 2, 2015. The frequency spectrum values were analysed for the Maule, Iquique and Illapel earthquakes.
Geophysical measurements are appropriate for highlighting fundamental frequencies; the frequencies generated in
these events ranged from 5.606 to 3.481 μ Hertz or from 1 cycle / 48.9 hours to 1 cycle / 79.13 hours.
In the Maule event, peaks for the frequencies 4.747; 5.064 and 5.154 μ Hertz were recorded [Figure 3 a]. In Iquique
peaks of 4.611; 4.882 and 5.154 μ Hertz were recorded [Figure 3 b]. And for Illapel 3.739; 4.630 and 5.520 μ Hertz
[Figure 3 c]. Before the Iquique 2014 event a jump in intensity was observed associated with the frequency of 5.154
μ Hertz [Figure 3 d] for the period Dec 27, 2013 to Jan 11, 2014, i.e. after the jump.  Figure 3 e shows a jump in
intensity associated with the frequency of 3.739 μ Hertz during Sep 1, 2015 to Sep 8, 2015 before the Illapel 2015
event and subsequent to the jump. Figure 3 shows a jump in intensity associated with the frequency of 3.739 μ Hertz
during the period Sep 1, 2015 to Sep 8, 2015, prior to the Illapel 2015 event but after the jump.
Generalizing the analyses illustrated, we consider the spectrograms of the second derivate of the $B_Z$ component (data
by minute) at the Easter Island station for the Maule 2010, Iquique 2014 and Illapel 2015 events [Figure 6a-b-
c][Rabiner and Schafer, 1978, Oppenheim et al., 1999]. In the Maule 2010 event, the spectrogram shows that the low
frequency behaviour around ~$0.01 - 0.3 \, mHz$ appears to diminish slightly in magnitude after the event of Feb 27,
2010 [Figure 6a]. The spectrogram for Iquique 2014 is marked with two arrows, one corresponding to Jan 3 and the
other (right) to the day of the event (Apr 1) [Figure 4b]. Within this period the magnitude of the minimum frequency
oscillated between $0.01 - 0.5 \, mHz$, however after the event the minimum frequencies tended to be around
~$0.5 \, mHz$, with occasional increases in frequencies close to ~$0.3 \, mHz$. The spectrogram for Illapel 2015 is shown in
Figure 4c, which presents the time period with two arrows: Jul 24 on the left, and Sep 16 (date of the event) on the
right. It can be seen that the low frequencies in the period start to increase progressively, from ~$1.2 \, mHz$ to
~$0.01 \, mHz$. After the event, the low frequency density appears to diminish, and the minimum frequency rises to
~$1 \, mHz$.





**4.  Daily cumulative numbers of anomalous behaviour in the component z of magnetic field over the surface of**
**Earth for Maule 2010 Mw8.8, Iquique 2014 Mw8.2 and Illapel 2015 Mw8.3**
In the method for cumulative magnetic anomaly in surface of earth, we used statistically an atypical or anomalous
value, that is, data that it is quite far from the average values of the sample. So we compare real values of $B_i$ with a
more representative value of the sample, its average $B_{ave.}$ We will call the difference between the two as the residual
$\Delta B$. Using the distribution of data we can define when a value is atypical or anomalous in a normal distribution by
statistical definitions of Quartiles and Outliers. The data used in this section comes from the supermag network
(http://supermag.jhuapl.edu/). The data has a sampling frequency of one data per minute, and a period of one year
before and one year after each earthquake was chosen.
We create a filter that eliminates the frequencies averaged near Nyquist and establishes a filter that eliminates high
frequencies. The option was to consider a moving average of five points weighted: $B_{ave} = aB_{i-2} + bB_{i-1} + cB_i + bB_{i+1} +$
$aB_{i+2}$. In our case we use a = 0.07, b = 0.25 and c = 0.5 − 2a. The uncertainty of the Flux-gate magnetometers used in
the OSO and PIL station is of $\delta B = \pm 0.1$ nT, which allows us to calculate the error propagation for averaged data $\delta B =$
$\delta B_i + \delta B_{ave} = \pm 0.2$ nT, then the total uncertainty is $\Delta B_i + 0.2$ nT. The data considered are for periods Dst <10 nT, and
only quiet magnetic data (6:00 - 05:00 Local time [Hitchmn et al., 1998]. $0.6745\sigma$ represents 50% of the data that is
closer to the average. So that 50% of the data farther from the average should be added. Then we consider as anomaly
$\delta Ba$ all the magnetic variations that are found at an amount $|\delta Ba|$ of the average value. If the threshold to define the
far points is 50% plus the error, then the equation to define the threshold or anomalies is $|\Delta Ba| >= 0.6745\sigma + 0.2$ nT.
To analyse the earthquake of Maule, we used the stations of OSO (Chile). The data used are between the 16.00 to
05.00 local time and in periods of time with an index DST less than or equal to 10 nT. OSO is a station nearby to be
approximately 450 km from the epicentre of the Earthquake Maule 2010. We use the $\Delta Ba$ equation, applied to the z
component of the OSO station. Its standard deviation obtained is 0.04020 nT (5 σ = 020104 nT), whereby the threshold
is $|\Delta Ba| >= 0.22712$ nT. The total anomalies registered for the two years are of 229, a normalized version is shown in
Figure 6. Between days Feb 27, 2009 and Jan 12, 2010 the anomalies register a linear behaviour. In the difference
between anomalies and linear interpolation for the period shows an increase in the approximate anomaly amounts
for three months (Apr 5, 2010). The Maule earthquake on Feb 2, 2010 (vertical red line), then starts another linear
period for approximately 8 months (Nov 27, 2010) with a new increase in the number of anomalies until the end of
the period, in this period there is a seismic swarm where some of these seismic movements are indicated with purple
double arrow [Figure 6]. A version without the linear trend is shown in Figure 7a.
In the earthquake of Iquique, we used Pilar station in direction North-East of Cordova in Argentina to 1420 Km of
epicentre (PIL). The data used are between the 16.00 to 05.00 local time and in periods of time with an index less
than or equal to 10 nT. The PIL station has almost no data between Dec. 27, 2014 and Feb. 3, 2015, however, this lack
of data will not affect further analysis as it is a relatively short period of time. It is also a period well after the Iquique
earthquake (9 months). The Z component of the PIL station is shown in Figure 7b. It has the values σ = 0.05262 nT (5σ
= 0.26310 nT), threshold $|0.23549|$ nT and a total of 165 anomalies. The interpolation takes place between Apr 1,
2013 and Oct 3, 2013. It presents three increases in the number of anomalies. The first on Oct 3, 2013, the second is
on Jan 8, 2014 and the third is on Apr 1, 2014. The change on Jan 8, 2014 is the most noticeable change in trend.
In the Illapel earthquake there was a great activity in the space environment, as the DST for 2015 is less precise, we
increased the requirement for the spatial filter. It was considered a DST less than or equal to 5 nT. The excess of
anomalies related with space weather was eliminated for 5 days in the PIL station: they are Apr 2, 2015; Sep 24, 2015;
Jan 31, 2016; May 16, 2016 and Aug 27, 2016. PIL is approximately 745 km from the epicentre, that is, a nearby
station. The $\Delta B$ equation, applied to the z component, its standard deviation obtained is 0.04797 nT (5 σ = 026803
nT) where by the threshold is $|\Delta Ba| >= 0.23615$ nT). The total anomalies registered for the two years are of 71.
Between days Sep 16, 2014 and May 3, 2015, the anomalies register a linear behaviour [Figure 7c]. There are two
increases in the number of anomalies prior to the Illapel earthquake, the first on Apr 28, 2015 and the second as of
Jul 18, 2015 being the most significant in the two years of records
The daily values of magnetic anomalies during a year before and one after the earthquake and the difference with
the linear interpolation have been made for the North and East components of all the stations studied, which we do
not include in this work, considering it redundancy.
Marchetti and Akhoondzadeh, (2018) referred to cumulative number of anomalous behaviour in the ionosphere,
which we have rebuilding, modified and plotted, with our method [Figure 8]. The first anomalies seen near 130 days
before the Mexico earthquake Mw8.2. They used in the study of cumulative numbers of anomalous track in
ionosphere with indices $|DST| < 20$ nT. For detected rapidly variation of magnetic field on Y component used the
method present in the Santis et al. (2017) (use of Cubic Splines instead of Moving Average.), and where indicates that





is easily affected by lithospheric activity. In general using satellite night time electron density residual variation date,
in pre-defined allowed ranges, based in M how median and IQR with inter quartile range parameters. M = + 1.25 x
IQR.
**5. Discussion and Conclusions**
The most significant characteristics of the total magnetic field and its variations in the first and second derivatives are
found in the $B_z$ component, which we observed and recorded in the Putre and IPM observatories. There is evidence
of a progressive increase in the phenomenon known as the South Atlantic Magnetic Anomaly (SAMA) Cordaro et al
(2019), generating greater deviation in the intensities present in the OP station [Figure 2 a] [Figure 2 b]. Combining
this information with data from the IPM station, the behaviour of the radial component of the geomagnetic field for
the three most significant seismic events in the Chilean Pacific sector during the period 2010-2015 was recorded and
it corroborates the magnetic relation with seismology shown by Potirakis et al. (2016), Contoyiannis et al. (2016) and
De Santis et al. (2017) using other methods. Furthermore, the same succession is observed in all the measurements
of this paper: Jump or second derivative of Bz, time lapse and seismic movement.
These jumps occur in different forms: in Putre they are significant, reaching values of tens of nT, while in IPM the
jump is barely 10nT. The time lapse between each jump and the seismic event differs in each event. For Maule 2010
it was 36 days, for Iquique 2014 it was 96 days, and for Illapel 16 days. This time difference may be due to an important
factor: it appears that the jump is not equally strong in the three events, since the jump before the Iquique 2014
event was considerably weaker than the one before Illapel 2015 [Figure 2], and preceded the event by a longer time
lapse (96 days). The more abrupt jump recorded in Illapel was followed by a shorter time lapse (16 days).
The secular variations are characterised by a peak on the day of the jump or the following day, while the secular
accelerations are characterised to a lesser degree by one positive and one negative peak. The frequency spectrum
found for each of the events in the second derivate of the geomagnetic field, or secular acceleration, demonstrate
the appearance of a range of frequencies between 3.5 and 5.5 µ Hertz [Figure 3 a, b and c]; there is a significant
increase in one or a group of frequencies specifically in the week of the jump [Figure 3 d, e]. This reflects the
oscillations of the radial magnetic field whose oscillation period takes from 2 to 4 days. We are therefore able to
propose a simple model for radial behaviour, associating a step function represented by the jump with a subsequent
oscillation represented by oscillations in a station in the field, e.g. Putre.
Figure 4 presents three spectrograms: All three show a reduction in the low frequency magnitude between $0.01 -$
$1\ mHz$ after the seismic events; the reductions for the Iquique and Illapel events are the most significant. Moreover,
for the Illapel 2015 event the spectrogram recorded a magnitude increase in the low frequency range $(0.01 - 1\ mHz)$
between Jul 24 and Sep 16; this increase was clearer than in the other events, possibly because the Illapel event
occurred at a similar latitude to the location of IPM (Illapel 2015: 31.573°S, IPM 27.171°S). The start of these increases
recorded in the spectrograms coincides with the dates identified as the jump in geomagnetic measurements for the
z component in the same station and in the frequency spectra obtained for the same time series [Figures 2 and 4].
Figure 5 shows the radial behaviour of the geomagnetic field at the Easter Island station for the Iquique 2014 event.
If this is compared with the records kept at the Putre station over the same period, we see that the gradients or trends
found in Putre and IPM are opposite; again, we see that the jump in IPM is of the order of 10 nT.
The records of the Putre station for the Illapel 2015 event [Figure 2c] show a change in the trend of the magnetic field
during the period from Sep 16 to Oct 8; this cannot be considered to be a time lapse, since the spectrogram (Figure
4c) shows no low frequency increase$(0.01 - 1\ mHz)$, there are only isolated increases lasting no more than one or
two days. The frequency range found for the second derivate of the radial component of the earth's magnetic field
$(0.01 - 1\ mHz)$ is within the typical range of the earth's free oscillations $(\sim0.1 - 1\ mHz)$ or of tidal effects $(\sim0.01 -$
$0.06\ mHz)$ [Casotto and Biscani, 2004, Park et al,. 2005], and is lower than De Santis et al. (2017) (~20mHz) due to
the fact that they use data from moving satellites and thus cannot cover lower frequency range. Furthermore, being
dependent on space weather, they must use few hours every night to record reliable data. Despite these technical
differences between satellites and ground level magnetometers used, the data show a time lapse between the start
of magnetic perturbations and earthquakes of order of one month (or even more) and it is similar to the time lapse
founded by De Santis et al. (2017).
In a complementary way, the magnetic anomalies method defined in section 4 was used. Figure 6 shows the daily
accumulated value of magnetic anomalies found for the vertical component at OSO station. The behavior of the
anomalies for the period of two years of data is relatively similar between the three events recorded at OSO and PIL
station (Figure 7). The three measurements show an initial stable period and a sudden increase in the number of daily
anomalies prior to the occurrence of the Maule, Iquique and Illapel earthquakes. Then a period appears again without
great variations. This described behavior is similar to that found by Marchetti and Akhoondzadeh (2018) for the



earthquake in Mexico using satellites (Figure 8). Figure 8 was carried out by applying a stricter spatial filter to that used by Marchetti and Akhoondzadeh (2018). This would indicate that the main source of anomalies could be in the lithosphere and not in outer space. It can be related to stress and electrification changes in rocks within the lithosphere (e.g. Tzanis and Vallianatos, 2002). However, the existence of a similar behavior in the anomalies recorded in the ionosphere suggests the existence of some coupling mechanism between the lithosphere, atmosphere and ionosphere.

We must repeat that we do not yet think that we can predict the future occurrence of these seismic events, since the seismological mechanism of seismic movements is not yet clear. However, a correlation does appear to exist between cumulative number of magnetic anomaly, jump, time lapse and seismic movement for Maule 2010, Iquique 2014 and Illapel 2015. This could be used as a tool to show the behaviour of some geophysical variables to indicate plate movements in the near but not immediate future. This condition, based on the increase of low frequencies($\mu Hz - mHz$), suggests that these magnetic variations in the radial component are probably a necessary but not sufficient condition on the Chilean margin; further investigation of this subject is required.

**Acknowledgments**

The authors thank Dr. E. Zesta (UCLA-IGPP), and Dr. L.A. Raggi (Incas-U.Chile) for their collaboration and support. We also thank our English–language consultant, Jorge Carroza. The fluxgate magnetometer at Putre-Incas Observatories is partially supported by FCFM-University of Chile, in collaboration with the South American Magnetometer B-Field Array (SAMBA/AMBER) project of the University of California, Los Angeles, USA, and Tarapaca University (Chile). LARC observatory The Chile/Italy Collaboration via U. Chile (Chile), PNRA (ITALY) supports, and INACh partial support. The results presented in this paper rely on data collected at magnetic observatories. We thank the national institutes that support them and INTERMAGNET for promoting high standards of magnetic observatory practice (www.intermagnet.org). D. Laroze acknowledges partial financial support from CONICYT- ANILLO ACT 1410, Yachay Tech startup, and Centers of excellence with BASAL/CONICYT financing, Grant FB0807, CEDENNA. E. Cordaro acknowledges Marcela Larenas and Francesca, Beatriz and Enrique for outstanding support to carry out this work

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



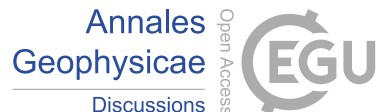

**Captions**
Figure 1: The Kp magnetic activity index for the periods prior to the Maule 2010 (top), Iquique 2014 (middle) and Illapel 2015
(bottom) earthquakes. [spidr NOAA] [WDCFG Kyoto University]
Figure 2: Component $B_z$ at the Putre station for a) Maule 2010 and b) Iquique 2014; c) Easter Island station for Illapel 2015
Figure 3: Fourier Fast Transform (FFT) of the $B''_z$ component in the Putre station for a) Maule 2010 and  b) Iquique 2014; c) Easter
Island for Illapel 2015; d) FT every 16 days for Iquique 2015 from the Putre magnetometer;  e) FFT every 8 days for Illapel 2015 from
the Easter Island magnetometer
Figure 4: Spectrograms with the behaviour of frequencies around 0.5 mHz, in periods close to the Maule 2010, Iquique 2014 and
Illapel 2015 seismic events
Figure 5: Trend for the Bz component to increase at Easter Island station after the jump on Dec 27, 2013 prior to the Iquique 2014
event.
Figure 6: The difference between anomalies and lineal interpolation in the period, for OSO Station in Maule event
Figure 7a, b, c: Accumulated Diary of magnetic anomalies during two years, in component Z, for Maule, Iquique and Illapel Events
Figure 8: Figure rebuild, modified, normalized and adapted with our method from Marchetti and Akhoondzadeh (2018).  Upper
panel: Accumulated Diary of magnetic anomalies during two years, in component Y from Apr 1 to Oct 15, 2017 in Mexico Earthquake
Sep 8, 2017 Mw8.2 and Lower panel: Residual behaviour of Mexico Earthquake.

Tables
Table 1. The main characteristics for the detector of Chilean network Cosmic Rays and Geomagnetic Observatories as location,
altitude, and atmospheric deept, type of detectors.
Table 2. The maximum radius where the ionosphere-lithosphere-atmosphere coupling may affect magnetic measurements to each
earthquake studied at the station of Putre and IPM [Dobrovolsky et al., 1979 ; Pulinets and Boyarchuk, 2004].



Figure 1

**Kp Index-Iquique 2014**

**Kp index-Maule 2010**

**Kp index -Illapel 2015**

10
11



Figura 2

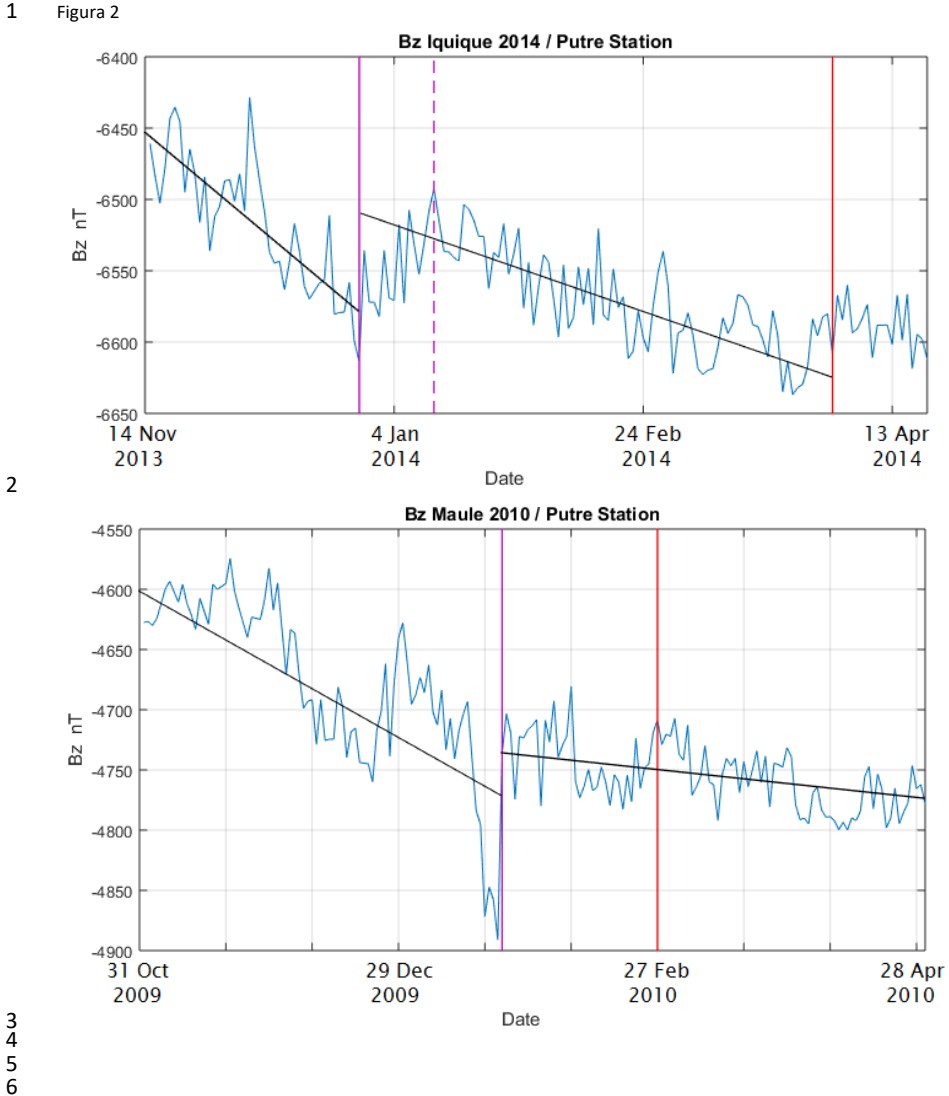




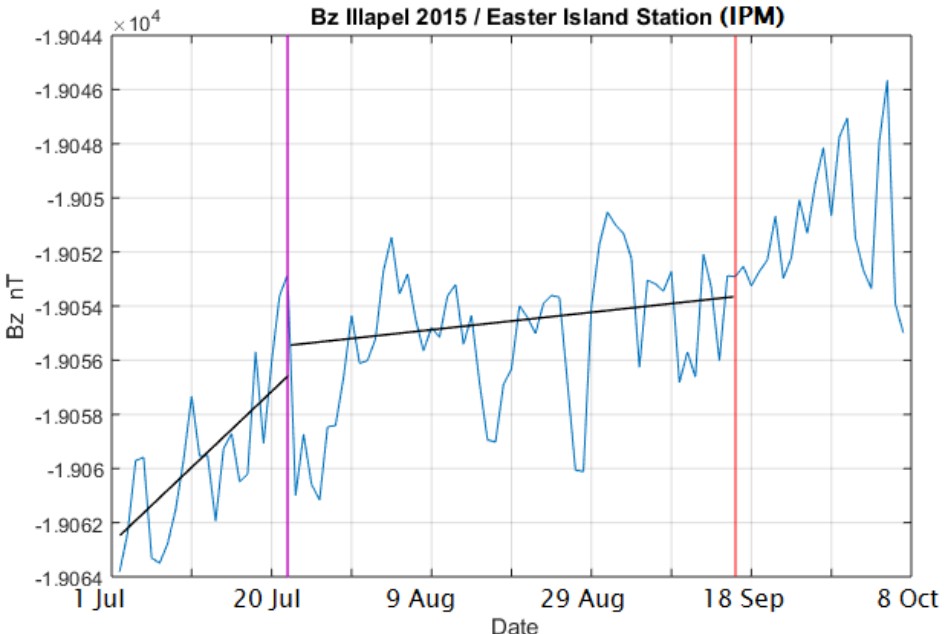

Figura 3

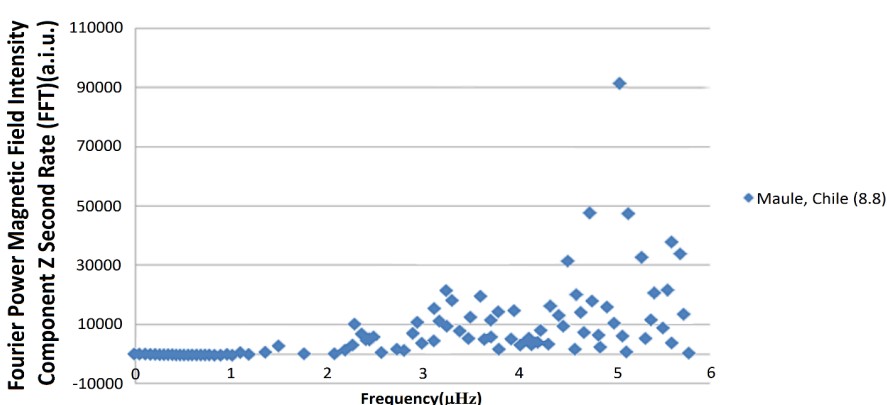




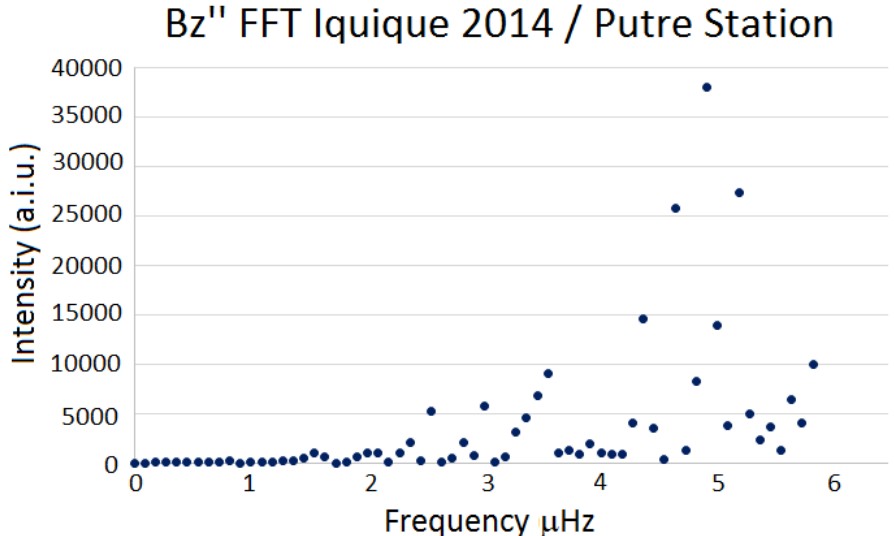

2

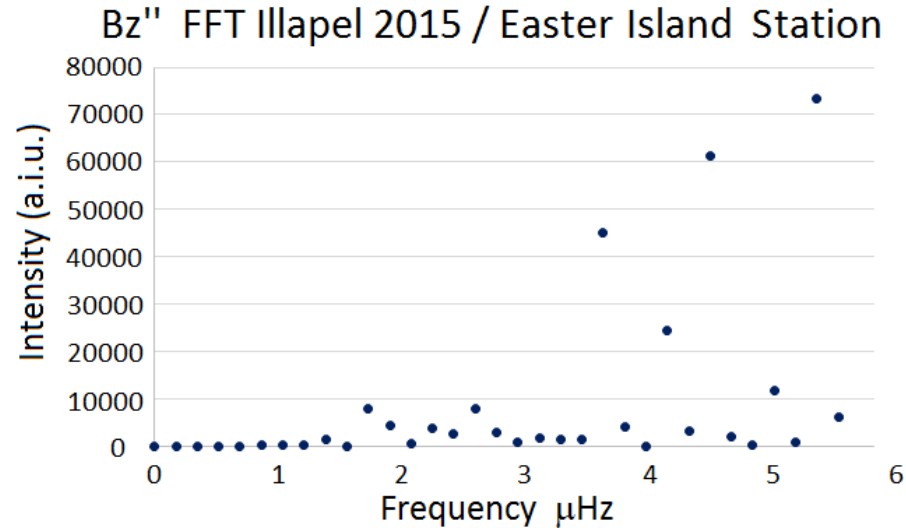

5





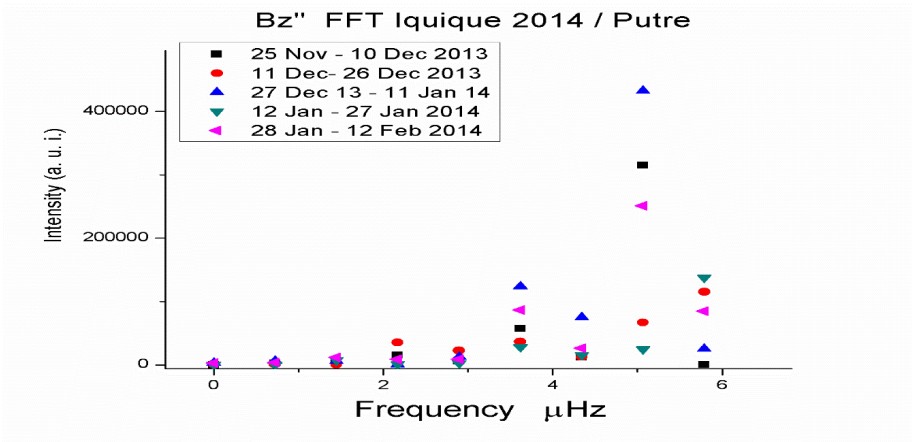

2

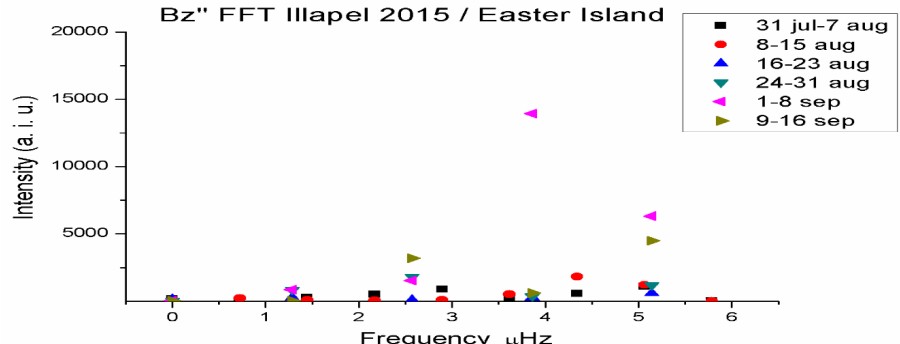

5
6

1    Figura 4

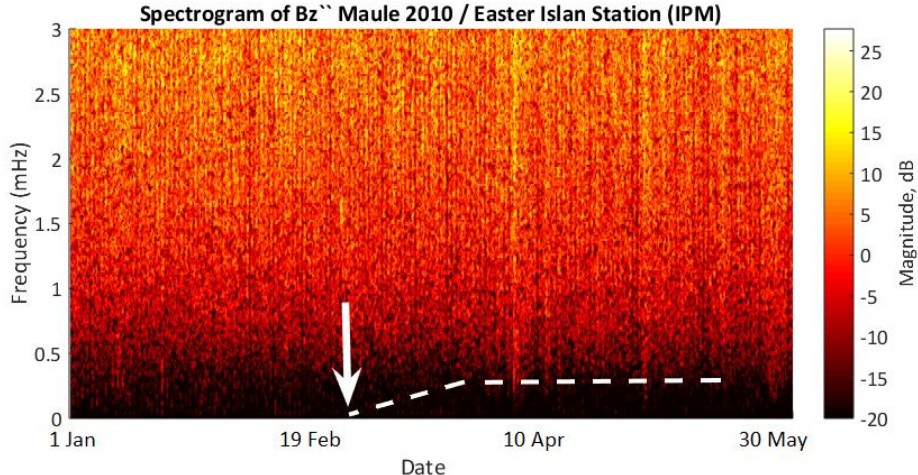

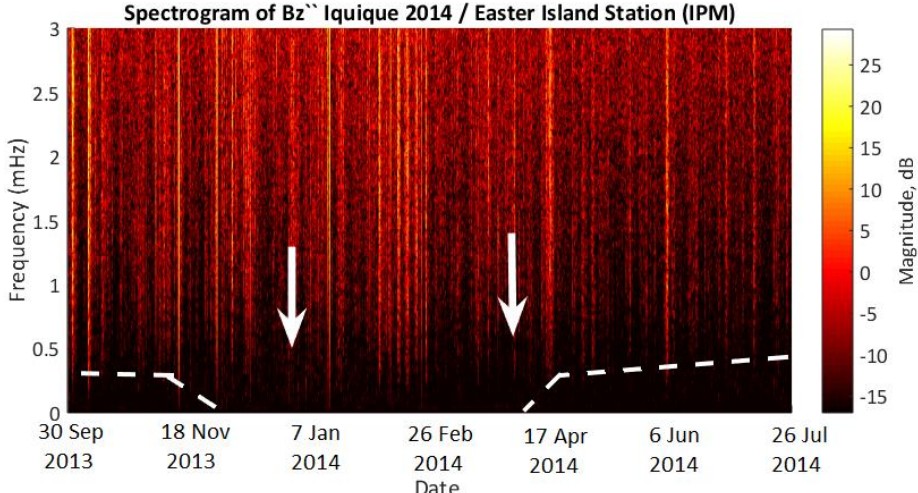



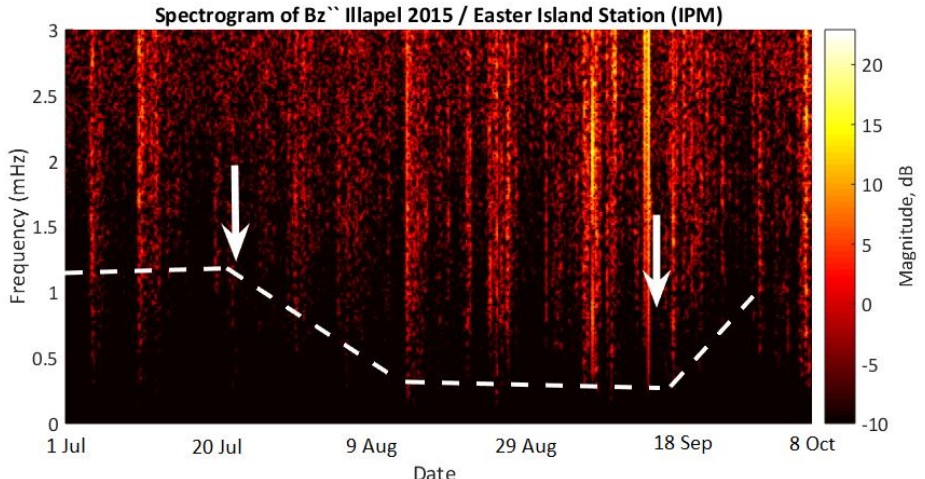

3    Figura 5

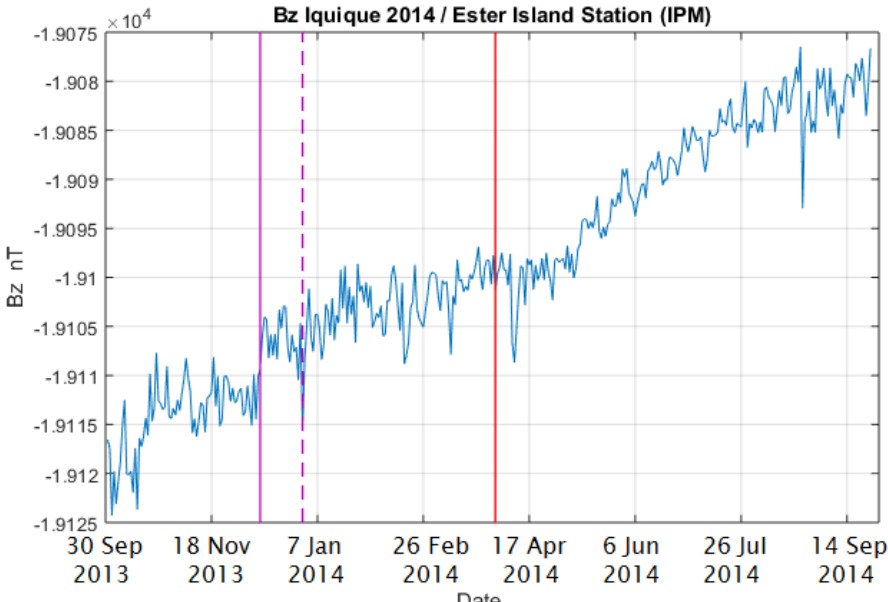





1    Figure 6

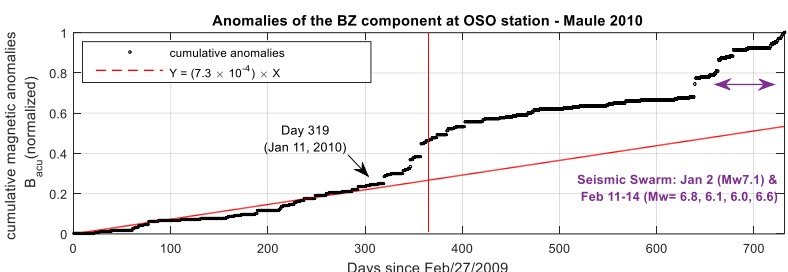

3    Figure 7

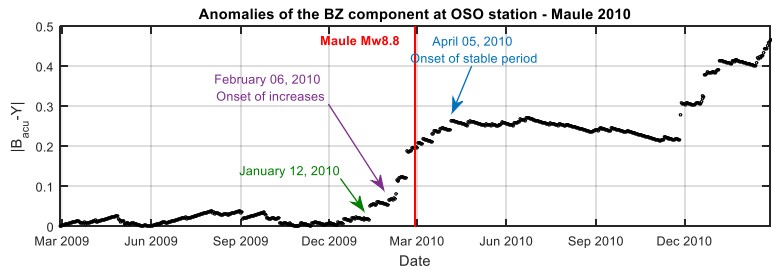

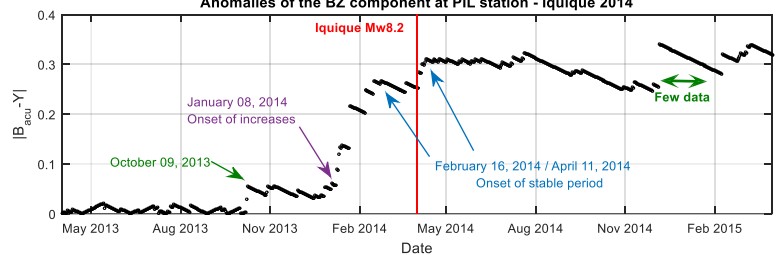

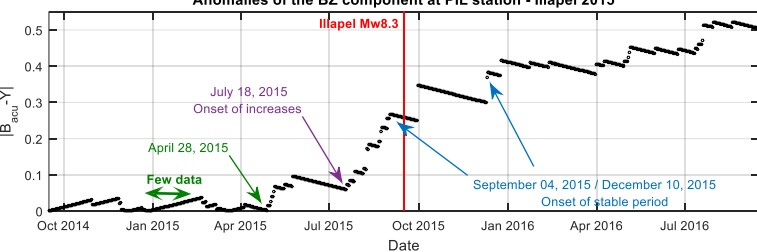



1    Figure 8

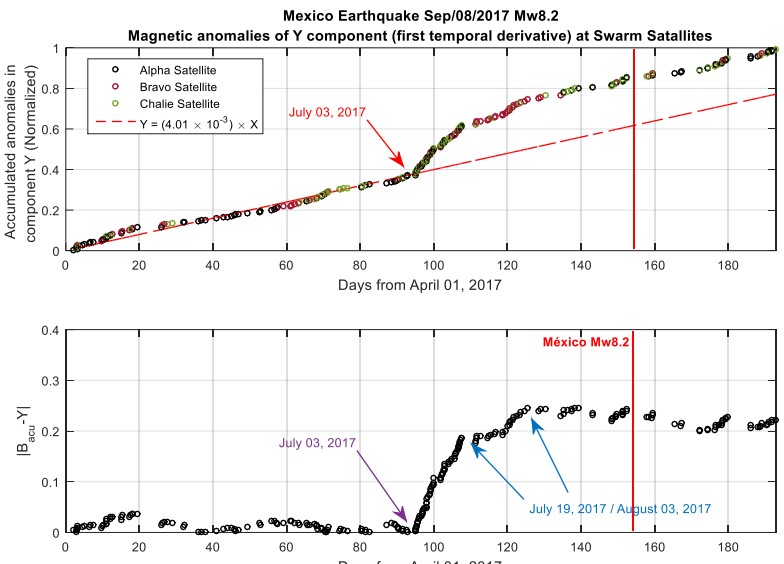

5    Table 1

| Observatory | Location | Geographical coordinate | Altitude [m.a.s.l] | Atmospheric Deep [g/cm2] | Instruments | Time |
|---|---|---|---|---|---|---|
| PUTRE (PUT) | Andes Mountain, Chile | 18°11´47.8 " S. 69°33´10.9" W | 3.600 | 666 | Magnetometer, UCLA-Vectorial-Flux Gate. Muon telescope, 3 channels. Neutron monitor IGY, 3 channels,He 3. UTC by GPS receiver. | 2003-2017 |
| Los Cerrillos (OLC) | Santiago de Chile, Chile | 33°29´42.2" S. 70°42´59.81 W | 570 | 955 | Magnetometer, UCLA-Vectorial-Flux Gate. Multi-dirctional muon telescope, 7 channels. Neutron monitor 6NM64, 3 channels, BF-3. UTC by GPS receiver. | 1958-2017 |
| LARC | King George Island, Antarctic | 62°12´9"S. 58°57´42¨ W | 40 | 980 | Magnetometer, UCLA-Vectorial-Flux Gate. Neutron monitor 6NM64 - BF-3BF-3. 6 channels. Neutron monitor 3NM64 – He-3. 3 channels, Neutron monitor 3NM64 – He-3.[Flux meter] 3 channels. UTC by GPS receiver. | 1990-2017 |

8    Table 2

| Event | Magnitude [Mw] | Radius r [km] | Station Distance from earthquake [km] |
|---|---|---|---|
| Maule 2010 | 8.8 | ~6100 | Putre ~ 2030 |
| Iquique 2014 | 8.2 | ~3360 | Putre ~ 300 |
| Illapel 2015 | 8.3 | ~3700 | IPM ~ 3700 |

