# Peer review of "Ann. Geophys. Discuss., https://doi.org/10.5194/angeo-2019-9 Manuscript under review for journal Ann. Geophys. Discussion started: 25 January 2019"

_Annales Geophysicae, 2019_

## Referee Comment (RC1) · Anonymous Referee #1 · 26 Feb 2019

The article "Analysis of geomagnetic measurements prior the Maule (2010), Iquique (2014) and Illapel (2015) earthquakes, in the Pacific Ocean sector of the Southern Hemisphere." by Cordaro, Venegas-Aravena and Laroze, is dealing with a very interesting topic, however the manuscript, in its present form, present several weaknesses. Apart from a number of typos or draft version leftovers here and there (e.g., "Ester Island", "Figura", "a (6:00 - 05:00 Local time...", etc.), one could highlight the following major issues: Part of section 2 is just a repeat of information given in introduction. The authors should carefully revise both section 1 & section 2 in the direction of a more

concise manuscript. The authors should use Z/Z* (e.g., doi: 10.1002/2014JA019789, doi: 10.1007/s10712-012-9215-x) or P_z/h (e.g., doi: 10.1002/2013JA019530, doi: 10.1080/19475705.2014.895965) instead of Z (or in parallel to Z), because from the literature these quantities seem to be most efficient in exposing precursory anomalies from ground-based magnetometers. The authors should also show the variation of Dst (probably also Ap) over the studied time periods. For all seismic events the authors should study the analyzed signals (e.g., Dst, Kp, B_z, Z/Z*, P_z/h, the derivatives of the magnetic field, etc.) over time periods of the same duration or explain why they don't do that. The readers would expect to have convincing information at hands in order to be persuaded that the revealed anomalies in the magnetic field recordings are not related to some global phenomenon. The whole part of Fourier analysis (both Fourier spectrum analysis and Spectrograms) is not convincing and probably should be excluded from the paper or carefully re-written. Some problematic points are the following: (a) it is not clearly stated which are the time periods over which each spectrum of Figs 3a-c was calculated; (b) for the peak at 5.154 uHz claimed to be related to the Iquique earthquake in Fig. 3d it is not clear why it presents intensity changes over the different studied time periods and what does this mean. (c) The claimed findings resulting from Fig. 4 are not supported by Fig. 4, while the white drawings (dashed lines and arrows) on the figures of Fig. 4 is not clear why have been employed (what are they highlighting). In section 4, the authors should explain why they adopted the specific filter and how did they ended-up with the used factors. Also, it is not clear how the anomalies were extracted from the measurements. The authors mention that "The data considered are for periods Dst <10 nT and only quiet magnetic data (6:00 - 05:00 Local time. . .". How did they selected the time period 16:00-05:00 as a quiet one and what do they mean by ". . .for periods of Dst <10 nT. . .", how precise in time is the specific discrimination, what happens around (but close to) the periods of Dst <10 nT? Finally, the authors have to bear in mind that there is recently a criticism regarding the formula proposed by Dobrovolsky et al. (1979) for the calculation of the region of precursory deformation, while the notion of critical radius is often used instead (e.g.,

doi:10.1002/2016JA023652, doi: 10.1002/2014JA019896, doi: 10.1785/0120040181, doi: 10.1029/98JB00792). Moreover, the original work of Dobrovolsky et al. (1979) itself indicates that some precursors may appear in narrower zones than others, while the wideness of the zone depends on the strain level (see Fig. 5 and Eq. (C) of Dobrovolsky et al., 1979). For example if the strain level is 10ˆ(-6), then an EQ of M8 corresponds to a radius of the order of 500 km and not of the order of 2700 km as calculated by the formula that the authors used. Therefore, their results may accept intense criticism, especially as regards the 2015 Illapel earthquake.

---

## Short Comment (SC1) · 12 Mar 2019

Patricio Venegas-Aravena

patricio.venegas@ing.uchile.cl

Dear referee # 1,

Thank you very much for your thoughts and suggestions. We have some typing errors that are easily solvable. It will not happen again. We consider the magnetic field Z and not Z / H, Z / Z *, etc ... since it is crucial to compare with other similar methods. For example 10.1016/j.epsl.2016.12.037 or https://doi.org/10.1016/j.asr.2018.04.043. In addition, we consider that these studies are much more solid since they allow to

observe the temporal evolution of the magnetic anomalies in periods of time of months or years. So it is much easier to observe if there is an increase in the number of anomalies on dates where earthquakes do not occur. It is important to emphasize that we only consider quiet times so it would not make much sense to show, for example Dst, in these conditions. In the manuscript we show that the spectrogram analysis is not enough, being of low quality in some cases and inconclusive. That is why we decided to focus on the behavior of anomalies. An important point that you comment is about the "globality" of this results. We could use distant magnetometers to corroborate if the behavior of the anomalies is similar to that registered in South America in dates close to large earthquakes. However, we have six years of total registrations in South America. During these six years several earthquakes of magnitude greater than, e.g., Mw7.8 (Nepal 2015 magnitud) occurred around the world. No behavior similar to that obtained for Chilean earthquakes was obtained, so this phenomenon cannot be global in nature. That's why we use the Dobrovolsky area as a reference. Although the deformations are not visible on the surface of the earth from a few hundred kilometers from the future epicenter, it does not necessarily imply that there are no changes of stress under the earth's surface at greater distances. For example, the generation of fractures occurs in the semi-fragile-ductile transition, between 10 and 20 kilometers below the earth's surface. Therefore, small changes in stresses or deformations at this depth may not be detected on the surface. A summary of the electrification in rocks in the non-elastic regime can be seen in 10.1016/j.pce.2003.12.003.

---

## Author Comment (AC1) · 1 Apr 2019

Dear Referee # 1

The referee makes explicit that the section devoted to Fourier analysis, both in Fourier spectrum analysis and spectrograms, is not convincing and suggests excluding it from the document or rewriting it. Based on this indication we will emphasize the concepts used and the data obtained. We present in Cordaro et al 2018 the variation rate of the geomagnetic stiffness cut between the years 1950 and 2010, this analysis we

found that the study of the vertical component of the magnetic field was necessary to study the seismic movements occurred in this area of the planet. Especially the behavior of the horizontal component, its first and second derivatives, before, during and after these events, their values and changes. we use as a method of analysis, the Fourier transform to calculate the frequencies for the events of Maule (2010), Sumatra (2004) and Tohoku Japan 2011., finding significant frequencies, which upon applying this method, we obtain for the events de Maule (2010), Iquique (2014) and Illapel (2015) presented in this paper, we would like to emphasize that in the Fourier Analysis of the second derivative of the Bz component we obtained the significant frequencies of the order of the microhertz prior to the seismic movements , corroborated by the low values of the Kp indices for the periods of seismic events in Iquique, Maule and Illapel. The periods of time considered that have not been explained in Figures 3 (a, b, c) have been considered and included.

In the analysis of spectrograms for the various events, they show reductions in the magnitudes of the low frequency between 0.01-1 mHz after the seismic events, indicating that the spectrogram recorded in Illapel presents an increase in the range of low frequencies between two dates. We have used daily averages for the period to be considered, moving these averages over a distance of several days in order to cover 80% of the data. Originally this analysis was carried out for the periods that included the seismic events of Maule (2010) Sumatra (2004) and Tohoku Japan (2011). Presented in the spectrogram of figure 4. In the detected and recorded data in the magnetometers in the observatories (Cosmic Radiation and Geomagnetism) located from equatorial to Antarctic zones along the Chilean coast show for each component of the magnetic field the changes of the intensity in each station during 3 or 4 hours after sunrise and 1 0 2 hours after sunset, even for those outside of SAMA (South Atlantic Magnetic Anomaly), which are the periods when the magnetic field is modulated by the traffic from day to night and night to day, also analyzing the increase in the rate of variation of particles of daytime cosmic radiation recorded in SAMA (Observation of intensity of cosmic rays and daily magnetic shift near meridian 70th in the Southemeric america EGCordaro etr

.. Journal of Atmospheric and Solar Terrrestrial Physics 142 (2016) 72-82) We have presented in this publication as an example for September 2008 the anisotropy of the compo This is the magnetic field for all the observatories of. Putre, Cerrilos and Antarctica, with a total range of variation from 0% to 100% where it is observed that between 14 and 16 hours UTC there is a maximum intensity of the magnetic field, We indicate that this phenomenon can be explained by means of the model for the magnetopause (Birkelan 1993, Russel te al, 1999) or attributable to periods of low Dst. It also includes the daily average of the intensity of the magnetic field for all the components (X, Y, Z) in 3D of the observatories of the Pacific Ocean sector of the Southern Hemisphere with data detected every minute and a scatter of the error every two hours.

Preferably in these comments we have privileged our vision of magnetic field strength and its importance in seismic movements. On the graphs of the values of magnetic anomalies before, during and after the seismic events indicating that the detected values are obtained in magnetometers on the surface of the earth, are grounds for other comments or other publication.

Enrique Cordaro on behalf of the authors.

---

## Referee Comment (RC2) · Anonymous Referee #2 · 3 Apr 2019

This paper focuses on an interesting science topic of the geomagnetic field variations prior to earthquakes. The authors analyzed the geomagnetic field records for three major earthquakes occurred in South America and showed geomagnetic field anomalies as earthquake precursors. I think the science results delivered by the paper does not meet the quality per the Journal requires. In addition, the text and figures contain many typos/errors, which makes them difficult to read.

My major concern is that the relation between the magnetic field variations shown and the earthquakes are not convincing. It might be that the magnetic field variations are

irrelevant to earthquakes, or the analysis method needs to be improved. The magnetic field data seems to be oscillating at all times, weak or strong. It is difficult to identify the changes prior and related to earthquakes if the black lines were not drawn in Figure 2 to guide eyes.

More importantly, the variability of the ground magnetometer data could be caused by many factors other than the earthquakes. One would argue that space weather induced impacts can be more direct and significant on the geomagnetic field. The authors made a fair attempt to show and analyze the Kp index, in order to rule out the variations due to space weather. However, the evidence is not convincing. Kp index is only one of many indicator of the geomagnetic activity. It is a global and daily index that may not reflect localized changes at times of interest. Therefore, more evidence is necessary to state "The magnetic records for the Bz component show little external influence." and "This would indicate that the main source of anomalies could be in the lithosphere and not in outer space".

Other comments include:

1. Section 1, what is the reason for the different frequency ranges for Fourier and spectrogram analysis; And what are the implications of the different frequencies?

2. Section 3, lines 18-22 on page 4: these do not read correct. The dates/years may be wrong.

3. In Section 4, the Dst index < 10nT could mean geomagnetic activity as it can go negative. I think the authors wanted to say the absolute value of Dst < 10 nT. In addition, what does it mean with "the DST for 2015 is less precise"?

4. The discussion and conclusion section should include limitations of the presented analysis and alternative explanations of the magnetic field anomalies if there is any. Moreover, what is the major contribution of this work?

5. The figure numbering does not match the main text and captions. All figure captions

need to be more descriptive and explain the legends on each figure. The dates on Figure 1 seems wrong.

6. It would be better to have a figure that shows the geographic locations of the earthquakes and the locations of the stations from which the geomagnetic data were obtained and analyzed.

---

## Author Comment (AC3) · 5 May 2019

Dear Natascha Töpfer and reviewers,

We have considered your suggestions and thoughts.

The changes made to the manuscript are highlighted in yellow. Below this letter you will find the answers to the referees.

Best Regards, E.G. Cordaro on behalf of the authors.

[Figure]

Comments for the Referee #2:

C1-" The text and figures contain many typos/errors, which makes them difficult to read."

R: To response this comment, the paper was corrected once more by native speakers.

C2-" The relation between the magnetic field variations shown and the earthquakes"

Page 4 , Lines 11-13

R: Different methods to search for the single cumulative magnetic anomalies and then the temporal behavior y now are used, in general in Ionosphere, for example: De Santis et al 2017. , Marchetti and Akhoondzadeh 2018 an other We used the method for cumulative magnetic anomaly in the surface of the earth.

C3-"Kp index is only one of many indicator of the geomagnetic activity."

R: we partially agree on this point, It's a global and daily index, considering this, we have indicated his maximum variation (figure 1) and the others source of anomalies are at the lithosphere, this objection in particular is to our understanding covered in the fourth section, "Daily cumulative numbers of anomalous behavior in the component z of magnetic field over the surface of Earth for Maule 2010 Mw8.8, Iquique 2014 Mw8.2 and Illapel 2015 Mw8.3".

C4- "For the different frequency ranges for Fourier"

Page 7, Lines 8-15

It is important to insist that the frequencies obtained by the Fourier method are inherent to the lithosphere, that is, obtained on the surface of the earth. The variation of the low frequencies prior to the Earthquake in the magnetic field are due to the ionosphere-atmosphere-lithosphere coupling. Other authors classify them in the ionosphere by relating them with waves transmitted from the magnetosphere and the solar wind. We indicate in Introduction (Cordaro et al (2018) that the frequencies in UHz are related

to Earthquake del Maule 2010 Mw 8.8 and Villanatos and Tzanis 2003 shows that the magnetic field frequencies are possibly related to Earthquake included in a range of at least three order of magnitude and finally detecting a month before the Earthquake in the range of frequencies between 5 -100 mHz based on the Ionosphere-Lithosphere –atmosphere coupling.

C5-" Lines 18-22 on page 4: these do not read correct"

R: It was checked again.

C6-"Think the authors wanted to say the absolute value of Dst < 10 nT. In addition, what does it mean with "the DST for 2015 is less precise"?"

R: That's correct, they are absolute values. And we want to say that the data is not as accurate.

C7-"What is the major contribution of this work?"

Page1 , Lines 39-40

R: The study of the afore mentioned variables could allow us to obtain precursor or pre-earthquake sings, and this may give us the possibility to predict an earthquake using measurable, objective readings. Allowing us to alert the population, so, the ultimate goal of this work is preserving life.

C8- "The figure numbering does not match the main text and captions. "

R: We checked, and accordingly highlighted the name of the figures in the main text and captions.

---

## Author Comment (AC4) · 5 May 2019

Dear Natascha Töpfer and reviewers,

We have considered your suggestions and thoughts.

The changes made to the manuscript are highlighted in yellow. Below this letter you will find the answers to the referees.

Best Regards, E.G. Cordaro on behalf of the authors.

[Figure]

Comments for the Referee #1:

C1-"Part of section 2 is just a repeat of information given in introduction"

R: We made a revision in section one and two to not repeat information given before. To be more explicit in section two, we do not repeat what we said in the introduction.

C2-"The author should use Z/Z*" Page 1 Lines 46-48

R: In this work we use field Bz or Z, and not Z/H, Z/Z* because it's how we call this component in past works. In the next work we considerate the suggestion made

C3-"Revealed anomalies in the magnetic field recordings are not related to some global phenomenon." and "the authors have to bear in mind that there is recently a criticism regarding the formula proposed by Dobrovolsky et al. (1979)"

Page 3, lines 48-55

R: To response this comment, In the globality of the results we could use distant magnetometers to corroborate if the behavior of the anomalies is similar to that registered in South America in dates close to large earthquake. However, we have six years of total registrations in South America, During these six years several earthquakes of magnitude greater than ,e.g, Mw 7.8 (Nepal 2015 magnitude) occurred around the world. No behavior similar that obtained for Chilean earthquakes was obtained , so this phenomenon cannot be global in natures That's why we use the Dobrosky area as a reference. Although the deformation are not visible on the surface of the earth from a few hundred kilometers from the future epicenter, it does not necessary imply that there are no changes of stress under the earth's surface at greater distance . (see in 10.1016/j.pce.2003.12.003.

C4: "it is not clearly stated which are the periods over which each spectrum of figura 3a-c was calculated"

R: To response this comment, the period of each spectrum on Fig3a-c were added.

C5: "For the peak at 5.154 uHz claimed to be related to the Iquique earthquake in Fig. 3d it is not clear why it presents intensity changes over the different studied time periods and what does this mean. "

Page 4, Lines 45-48

R: The changes of intensity require more studies to understand what they mean. This frequencies and numbers are what nature give to us, therefore measurements, so Geophysical measurements obtained are appropriate for highlighting fundamental frequencies at the Iquique earthquake.

C6: "What happens around (but close to) the periods of Dst <10 nT"

Page 5, Lines 33-35

R: Disturbance storm time , computed in 4 mid-latitude observatories to obtain the average measurement of magnetic field variations, this allows us to detectet variations of magnetic field and magnetic storms when they occur) In this case is absolute values of DST